# The burden of chronic diseases among Australian cancer patients: Evidence from a longitudinal exploration, 2007-2017

Rashidul Alam Mahumud[1,2,3,4]*, Khorshed Alam[1,2], Jeff Dunn[1,5,6], Jeff Gow[1,2,7]

**1** Health Economics and Policy Research, Centre for Health, Informatics and Economic Research, University of Southern Queensland, Toowoomba, Queensland, Australia, **2** School of Commerce, University of Southern Queensland, Toowoomba, Queensland, Australia, **3** Health Economics Research, Health Systems and Population Studies Division, International Centre for Diarrhoeal Disease Research, Bangladesh (icddr,b), Dhaka, Bangladesh, **4** Health and Epidemiology Research, Department of Statistics, Rajshahi, Bangladesh, **5** Cancer Research Centre, Cancer Council Queensland, Fortitude Valley, Queensland, Australia, **6** Prostate Cancer Foundation of Australia, St Leonards, New South Wales, Australia, **7** School of Accounting, Economics and Finance, University of KwaZulu-Natal, Durban, South Africa

* rashed.mahumud@usq.edu.au, rashidul.icddrb@gmail.com

**Data Availability Statement:** This paper uses unit record data from the Household, Income and Labor Dynamics in Australia (HILDA) Survey under

## Abstract

### Introduction

Cancer is a major public health concern in terms of morbidity and mortality worldwide. Several types of cancer patients suffer from chronic comorbid conditions that are a major clinical challenge for treatment and cancer management. The main objective of this study was to investigate the distribution of the burden of chronic comorbid conditions and associated predictors among cancer patients in Australia over the period of 2007–2017.

### Methods

The study employed a prospective longitudinal design using data from the Household, Income and Labour Dynamics in Australia survey. The number of chronic comorbid conditions was measured for each respondent. The longitudinal effect was captured using a fixed-effect negative binomial regression model, which predicted the potential factors that played a significant role in the occurrence of chronic comorbid conditions.

### Results

Sixty-one percent of cancer patients experienced at least one chronic disease over the period, and 21% of patients experienced three or more chronic diseases. Age (>65 years old) (incidence rate ratio, $IRR$ = 1.15; 95% confidence interval, $CI$: 1.05, 1.40), inadequate levels of physical activity ($IRR$ = 1.25; 95% CI: 1.09, 1.59), patients who suffered from extreme health burden ($IRR$ = 2.30; 95% CI: 1.73, 3.05) or moderate health burden ($IRR$ = 1.90; 95% CI: 1.45, 2.48), and patients living in the poorest households ($IRR$ = 1.21; 95% CI: 1.11, 1.29) were significant predictors associated with a higher risk of chronic comorbid conditions.

strict licensing. Although data are not available to the public, they can be potentially obtained and shared subject to a peer-reviewed application. The data are available from the Australian Government Department of Social Services and the Melbourne Institute of Applied Economic and Social Research at https://melbourneinstitute.unimelb.edu.au/hilda.

**Funding:** The authors received no specific funding for this work.

**Competing interests:** The authors have declared that no competing interests exist.

## Conclusions

A large number of cancer patients experience an extreme burden of chronic comorbid conditions and the different dimensions of these in cancer survivors have the potential to affect the trajectory of their cancer burden. It is also significant for health care providers, including physical therapists and oncologists, who must manage the unique problems that challenge this population and who should advocate for prevention and evidence-based interventions.

## Introduction

Cancer is one of the most pressing public health problems worldwide [1]; an estimated 9.6 million patients die from cancer each year. In Australia, it is also an alarming issue with the health system dealing with 483 new cases per 100,000 people in 2019, while on average 136 people die from cancer each day [2]. Cancer contributes 18% of the total burden of disease in terms of disability-adjusted life years, followed by 14% from cardiovascular diseases, 13% from musculoskeletal conditions, and 12% from mental and substance use disorders in Australia [3]. Further, there are approximately one million survivors in Australia who have been diagnosed with cancer in the past [4]. The five-year survival from all cancers combined improved from 48% to 69% between 1990 and 2011–2015 [2].

However, the majority of cancer patients suffer from chronic diseases or conditions, commonly referred to as comorbidity. The risk of having comorbidity increases during treatment as well as oncology follow-up periods [3,5,6], which adversely influences treatment choices and outcomes. Chronic comorbid conditions of cancer patients contribute to a major clinical challenge in terms of cancer diagnosis, ill health, the course of treatment, long-term disability and disease management [7]. In 2014–15, more than 11 million Australians (50%) reported having at least one chronic disease, wherein approximately 1 in 4 (23%) Australians had two or more chronic conditions [8]. This rate was more pronounced for people aged 65 and over (87%) compared with people aged 0–44 (35%), females (52%) compared with males (48%), people in disadvantaged socioeconomic areas (55%) compared with those in the most advantaged socioeconomic areas (47%), and people living in regional and remote areas (54%) compared with those in the major cities (48%) [8]. Ultimately, the severity of comorbidity leads to an increased risk of hospitalisation, reduced health status, increased mortality, and increased financial burden on the healthcare system [9–11]. It may also adversely impact an individual's access to advanced cancer treatments (e.g., chemotherapy and radiotherapy) and the effectiveness of that treatment [12]. This is a substantial prognostic factor for the long-term survival of cancer patients. There is a growing body of research on the significant impact of chronic comorbid conditions among patients with cancer. However, there are limited empirical studies on comorbidities available in the Australian setting [7,13–15].

Comorbidity has a well documented detrimental effect on cancer survival [9] and it describes the existence of a long-term health condition or disorder in the presence of primary disease or illness [16]. In the case of cancer, chronic comorbidity refers to the existence of one or more comorbid conditions in a person simultaneously. While the existence of these comorbid health conditions may be extraneous, particularly chronic diseases, there is an association between them. Further, many chronic diseases share common risk factors. Cancer patients with comorbid conditions also experience a higher physiological burden of disease [7]. The presence of specific severe comorbidities or psychiatric disorders is associated with delayed

cancer diagnosis [11]. Further, patients with chronic diseases with regular medical consultations and follow-up had their cancer detected at an earlier stage [12].

The chance of improving health status and completing a course of cancer treatment in the presence of comorbidities is significantly lower among cancer patients [4,13,15,17,18] and is associated with a higher rate of mortality depending on the severity of disease and associated comorbidity [11]. For instance, the mortality rate is substantially higher among cancer patients with comorbidities (47%) compared with cancer patients without comorbidities (34%) [19]. Given the clinical significance of comorbidity and its high prevalence in cancer survivors, it is essential to have a measure for quantifying likely effects on cancer outcomes [20]. Understanding more about comorbidities among cancer patients can generate possible evidence as well as provide direction for prevention, management, and treatment of chronic diseases.

A number of studies confirm that comorbid chronic conditions were more pronounced among cancer patients [4,11,13–15,21,22]. The most prevalent risk factors were age (over 65 years) [23,24], unhealthy behaviors (e.g., alcohol consumption and smoking tobacco) [25,26], obesity, limited engagement with physical activity [27] and inadequate diet [25] and they are significantly related to a higher risk of developing cancer along with multiple chronic diseases [5,7,25]. Further, comorbid conditions of cancer patients are significantly associated with worse health status during treatment and oncology follow-up periods [28,29] as well as low or intermediate socioeconomic status [30], and poor nutritional status [31]. The ongoing evidence shows that modifying or avoiding risk factors can significantly reduce the burden of chronic comorbid conditions among cancer patients [1]. For example, cancer survivors who engage in less sedentary behavior enjoy a better quality of life [32], and this can also significantly contribute to reducing the risk of experiencing chronic comorbid conditions [33].

The primary intention of these studies was to examine the distribution, trend, pattern, and disparity in comorbidity status among cancer patients when considering a limited range of variables. The majority of these studies pay little attention to examining the long-term impact of chronic comorbid conditions for cancer survivors' over times. Therefore, routine oncology follow-ups must explore how cancer survivors' characteristics impact on the number of chronic comorbid conditions they experience.

This study will examine the longitudinal nature of chronic comorbid conditions of cancer patients. More specifically, the study proposes to develop a better understanding of the longitudinal distribution of chronic comorbidity status among cancer patients as well as its impact over time. This study complements and contributes to this strand of ongoing cancer research to increase awareness and improve public health practice among sufferers and survivors, and to measure impact. The findings could contribute to designing appropriate interventions and/or the provision of quality healthcare services and resources for ongoing surveillance of people living with, through and beyond cancer, and help determine what kinds of support survivors need. This study, therefore, aims to investigate the distribution, potential predictors and associated burden of chronic comorbid conditions among cancer patients by using a longitudinal data set from the Household, Income and Labour Dynamics in Australia (*HILDA*) survey.

## Materials and methods

### Study design

The study design is a longitudinal exploration using a household-based panel over an extended period of 2007 to 2017. Individuals who face the burden of life-threatening cancer were interviewed with a focus on the magnitude of the cancer burden associated with their chronic comorbid conditions. The magnitude of the cancer burden includes their course of treatment

over an extended oncology follow-up period which can affect their health status burden and includes chronic comorbid conditions, disability, and adverse events.

## Conceptual framework

The distribution of comorbidity varies by patient-level factors (Fig 1). Like cancer itself, it increases with age. Functional status, a measure of patients' ability to perform everyday activities, is related to both the presence and the consequences of chronic comorbid conditions. Health status burden is associated with increased vulnerability to stressors that result from decreased health scores as well as physiological strength [34]. Further, health status burden is strongly associated with increased age and the severity of the disease. In the context of comorbidity experiences, patients assess their health status depending on the severity of disease (as either better or worse) [35]. Despite strong associations between them, comorbidity, functional status, and health status burden are separate entities, and each has an independent effect on outcomes [34]. To investigate the longitudinal effects, it is assumed that several predictors (e.g., individual background characteristics, social factors, and disease-related symptomatic factors), measured at the symptom-level might predict outcome factors (e.g., appraisal of disease severity levels, utilisation of advanced treatment, life satisfaction, and uncertainty). Moreover, the combination of predictors was expected to predict patients' health outcomes (e.g., chronic comorbid conditions, long-term health problems or disability, and adverse events).

## Data source

Data came from the Household, Income and Labour Dynamics in Australia (*HILDA*) survey [36]. The *HILDA* survey commenced in 2001 and is a nationally representative household-based panel study that produces data on the lives of Australian residents aged 15 or over. As per the *HILDA* protocol, written or verbal consent was collected from all potential participants before conducting the survey. Data were collected through face-to-face interviews using quantitative survey instruments, followed by re-interviews with the same people in subsequent years. The details of the methods of data collection, including the sampling technique, have been explained elsewhere [36]. The present study participants were diagnosed with cancer patients,

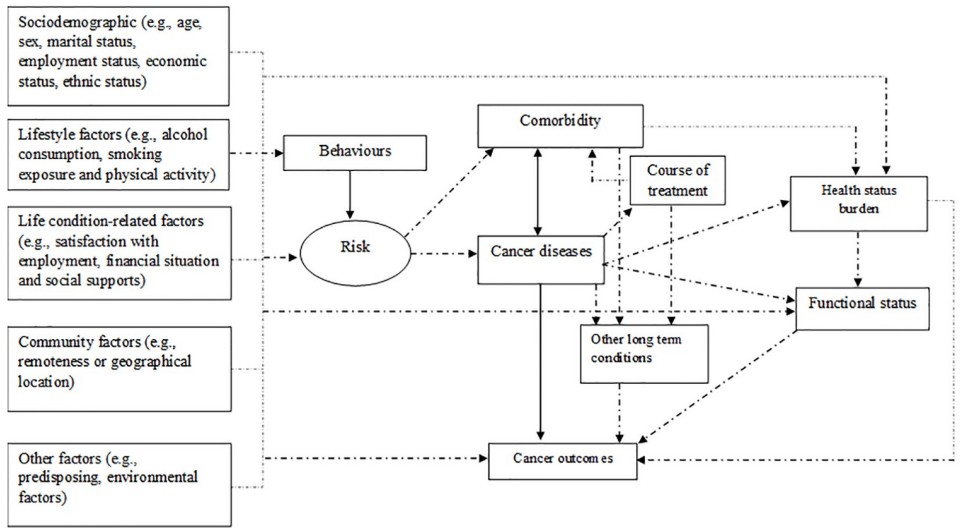

**Fig 1. Conceptual framework of the study.**

and data were restricted to four waves (e.g., wave-7, wave-9, wave-13 and wave-17) based on the availability of data related to cancer. However, wave-3 was excluded from the analysis due to the limited data related to comorbidity status. Other survey waves were excluded from the analyses due to the paucity of cancer-related information. A total of 2,066 diagnosed cancer patients were potential study participants from the four waves: wave-7 in 2007 ($n_2$ = 557), wave-9 in 2009 ($n_3$ = 416), wave-13 in 2013 ($n_4$ = 517) and wave-17 in 2017 ($n_5$ = 576).

## Study variables

**Outcome variable.** The chronic comorbid conditions were classified into disease groupings and cover the most common types of long-term health conditions experienced by cancer patients in the Australian community. A previous review study identified that at least 21 approaches have been executed to measure comorbidity status [37]. There is no gold-standard method for measuring comorbidity among cancer populations [37]. The selection of the method depends on the study research question, data availability, and population studied. A number of methods related to measuring comorbidity status have been used in the context of cancer-related studies including exploration of the impact of single conditions (such as diabetes or congestive heart failure) [38–40], single condition counts [41–43], weighted indices [43–47], and organ-based systems [48–50]. Although all these approaches aim to evaluate the same underlying construct, they vary in terms of the study purpose for which the measures were performed. These approaches vary in the context of study perspective and design. The simplest approach to measuring comorbidity status is to investigate the distribution of individual comorbid conditions and to treat them independently and/or to combine them by summing the total number of conditions [51]. In this study, a single condition count approach was performed to measure comorbidity status. Cancer patients reporting chronic condition(s) were considered an outcome variable in the analysis. Chronic comorbid conditions included being diagnosed with serious chronic illness, including arthritis or osteoporosis, heart disease, diabetes, hypertension, mental illness, or circulatory conditions. The count of chronic health conditions was measured for each respondent based on the number of disease exposures and who had been prescribed medication for their illness. If the respondents had multiple chronic conditions, it was counted as multiple responses.

**Explanatory variables.** This study considered several demographic, socio-economic and health and lifestyle-related variables based on the conceptual framework, as putative predictors of chronic comorbid conditions. Socio-demographic factors, such as sex, age, educational achievement, employment status, and marital status were considered as potential factors in the analysis. Lifestyle factors such as alcohol consumption, smoking exposure, and physical activity were also included. The level of physical activity was categorized into three groups as low, moderate, or high [27,52,53]. Further, life condition-related factors such as satisfaction with employment, financial situation, and social supports were also selected as potential predictors. Ethnic status was defined as Aboriginal or non-Aboriginal. The quality of life scores was measured using the medical outcomes study short-form (*SF-36*) [54]. The SF-36 is one of the most common generic measures of health-related quality of life, which is widely used to assess the burden of disease in the context of different country settings [55]. It uses psychometric properties to enable profiling of physical functional health and well-being and to quantify disease burden across eight domains, including physical functioning, role-physical, body pain, general health, vitality, social functioning, role-emotional, and mental health. Considering these dimensions, the total score on each *SF-36* subscale ranges between 0 and 100, labelling 'worst imaginable health' and 'best imaginable health state', respectively. It is signified that the higher scores represent better health status. A recent review study confirmed that several studies used a total score of *SF-36* items to derive quality of life scores across the eight domains of *SF-36*

[56]. The levels of health status burden were proposed based on the magnitude of quality of life scores as follows: (1) high burden if the short form-36 (*SF-36*) scores < 50.00, (2) moderate burden if 50.00 ≤*SF-36* scores < 90.00, and (3) no burden if *SF-36* scores ≥ 90.00. The level of health status burden captured the severity of disease for cancer patients. Work disability was measured based on the severity of disability score ranged from 0 to 10, with 10 indicating 'able to do any work' and 0 indicating 'not at all'. The severity of disability level was defined as follows: (i) 'no disability' if disability score was equal to zero, (ii) 'moderate disability' for disability scores of 1 to 6, and (iii) 'severe disability' for disability scores of 7 to 10. Geographical locations were defined according to the accessibility to services and the Remoteness Index of Australia [57], and they were categorized into five groups: major cities, inner regional, outer regional and remote or very remote. The index of relative socioeconomic disadvantage (*IRSD*) was used to measure socioeconomic status (*SES*). The index was defined into five groups with these threshold values: $Q_1$ (*IRSD* ≤ 927.0), $Q_2$ (927.0 > IRSD ≤ 965.8), $Q_3$ (965.8 > *IRSD* ≤ 1001.8), $Q_4$ (1001.8 > *IRSD* ≤ 1056.0), or $Q_5$ (*IRSD* > 1056.0) [58]. This is a geographical area-based estimate of socioeconomic status using income, education level and occupation where communities are categorised from economically disadvantaged to wealthy.

## Statistical analysis

This study utilised descriptive analyses to compare patients with cancer and chronic medical conditions across the characteristics. The trend of chronic comorbid conditions among cancer patients was performed using the Cochran-Armitage trend test [59]. In the analytical exploration, the adjusted fixed-effect negative binomial regression model was used to identify the potential factors that had a significant role in the exposure to chronic comorbid conditions. In the regression model, the dependent variable (number of chronic comorbid conditions) was characterised as a count measure. An unadjusted analysis was performed using only separated explanatory variables for the following reasons: (1) primary screening of the selection of qualified predictors, which were added in the adjusted model, (2) although the chi-square tests (or one-way analysis where appropriate) are only used to find the association between outcome and explanatory variables. However, the majority of the predictor variables were categorical nature with two or more labels in this study. Therefore, an un-adjusted analysis was performed to find the association between outcome and the labels of explanatory variables. The predictor variables were included in the adjusted model only if any label of the predictor was significant at 5% or less risk level in the unadjusted model, which in turn was used to adjust for the effects of other potential confounders. However, insignificant predictors were not included in the adjusted model. The model was tested for sensitivity by the forward selection procedure (e.g., including and excluding specific variables) with robust standard errors. For the independent variables, the category found to be least at risk of having chronic comorbid conditions in the analysis was considered as the reference for constructing incidence risk ratios (*IRR*). Statistical significance was considered at the 5% risk level. All data analyses were undertaken using the statistical software Stata/SE 13 (StataCorp, College Station, TX, USA).

## Ethical considerations

The Household, Income and Labour Dynamics in Australia (*HILDA*) data are used under strict licensing. Data can be potentially obtained and shared subject to a peer-reviewed application. Ethical approval for the *HILDA* study was obtained from the Faculty of Business and Economics Human Ethics Advisory Committee at the University of Melbourne (#1647030). Approval for the use of *HILDA* data was provided by the Department of Social Services. Ethical approval was not required from an institutional review board because the patient information

was de-identified. Appropriate approval was obtained for this study from the Department of Social Services to access the de-identified longitudinal dataset.

# Results

## Background characteristics of the study population

A total of 2,066 cancer patients were potential participants (Tables 1 and 2). Approximately 54% of patients were male, with 58% of patients being married. A higher proportion (46%) of the patients were senior or old senior-aged (more than 65 years), followed by middle-aged (37%). Approximately 47% had completed middle or high school level education, with 316 cancer patients (15%) having tertiary education. Sixty three percent of 63% of patients were unemployed, while 45% of patients had inadequate physical activity, with only 23% of patients having high-level physical activities per week. Two-third of 75% of patients consumed alcohol frequently. The majority of participants (89%) reported a moderate or extreme health burden, whereas 42% of patients experienced moderate or severe disability levels. In addition, 72% received prescribed medication, and 61% lived in major cities.

## Distribution and changes of chronic comorbid conditions with cancer patients over time

The prevalence of comorbid conditions was reported by cancer patients as follows: arthritis or osteoporosis (45%), high blood pressure or hypertension (39%), obesity (23%), depression or anxiety (22%), heart disease (14%), and asthma (13%). These were significantly increased in the prevalence of depression or anxiety (p<0.01), mental illness (p = 0.052) and obesity (p = 0.003) over the period (Fig 2). However, a downward trend in the prevalence of comorbid conditions was observed for arthritis/osteoporosis (p = 0.012) over time.

Overall, approximately 42% of patients suffered from one to two chronic comorbid conditions, while 21% of patients experienced at least three or more comorbid conditions (Table 1). The prevalence of comorbid conditions was prominently distributed by age. The majority of comorbidities were highly pronounced in patients due to a lack of physical activity. For example, 56% of patients were more likely to report three or more comorbid conditions. This prevalence was disproportionately low (14%) in those who engaged in a high level of physical activity. Further, patients who suffered from at least one comorbid condition were significantly aligned with the magnitude of high or moderate health status burden (e.g., 62% for severe burden and 36% for moderate burden). Similarly, an upward trend of the upper extremity of disability levels was observed with an increased number of comorbid exposures among the poorest cancer survivors during the period (Fig 3). Regarding socioeconomic position, the magnitude of comorbid conditions was more pronounced in the most disadvantaged socio-economic group. For example, 28% of patients who lived in the poorest households were significantly exposed to three or more comorbid conditions compared with the richest households (13%). Also, the severity of disability score was also highest among patients in the poorest households along with an increasing number of comorbid conditions (Fig 3).

## Factors influencing chronic comorbid exposure of cancer patients

Table 3 exhibits the results of the fixed effect negative binomial regression analyses. In the adjusted model, older patients, the magnitude of health status burden associated with cancer, utilisation of healthcare, and patients living in the poorest households were significant predictors associated with a higher risk of comorbid conditions. An aged patient (>65 years old) has 1.15 times higher risk of having comorbid conditions (incidence rate ratio, $IRR$ = 1.15; 95%

**Table 1. Summary statistics by the number of chronic condition among cancer patients for wave 7 and wave 9.**

| Variables | Number of observations, n (%) | Wave-7 Number of chronic comorbid conditions, n (%) | | | Wave-9 Number of chronic comorbid conditions, n (%) | | |
|---|---|---|---|---|---|---|---|
| | | 0 | 1–2 | 3 or more | 0 | 1–2 | 3 or more |
| Sex | | | | | | | |
| Male | 1,123 (54.36) | 234 (54.80) | 77 (59.23) | na | 45 (51.14) | 110 (48.46) | 55 (54.46) |
| Female | 943 (45.64) | 193 (45.20) | 53 (40.77) | na | 43 (48.86) | 117 (51.54) | 46 (45.54) |
| Age | | | | | | | |
| <25 years | 53 (2.57) | 10 (2.34) | 3 (2.31) | na | 1 (1.14) | 4 (1.76) | 1 (0.99) |
| 25–45 years | 283 (13.70) | 77 (18.03) | 17 (13.08) | na | 23 (26.14) | 32 (14.10) | 8 (7.92) |
| 46–65 years | 771 (37.32) | 146 (34.19) | 69 (53.08) | na | 39 (44.32) | 86 (37.89) | 30 (29.7) |
| >65 years | 959 (46.42) | 194 (45.43) | 41 (31.54) | na | 25 (28.41) | 105 (46.26) | 62 (61.39) |
| Educational attainment | | | | | | | |
| Year 11 or below | 774 (37.46) | 169 (39.58) | 48 (36.92) | na | 26 (29.55) | 97 (42.73) | 46 (45.54) |
| Year 12 | 168 (8.13) | 37 (8.67) | 14 (10.77) | na | 10 (11.36) | 15 (6.61) | 9 (8.91) |
| Trade/certificate/diploma | 808 (39.11) | 149 (34.89) | 54 (41.54) | na | 35 (39.77) | 81 (35.68) | 40 (39.6) |
| Tertiary | 316 (15.30) | 72 (16.86) | 14 (10.77) | na | 17 (19.32) | 34 (14.98) | 6 (5.94) |
| Unemployed | 1,306 (63.21) | 250 (58.55) | 66 (50.77) | na | 40 (45.45) | 150 (66.08) | 86 (85.15) |
| Marital status | | | | | | | |
| Single | 258 (12.49) | 52 (12.18) | 20 (15.38) | na | 16 (18.18) | 27 (11.89) | 7 (6.93) |
| Married | 1,196 (57.89) | 256 (59.95) | 80 (61.54) | na | 46 (52.27) | 130 (57.27) | 52 (51.49) |
| Others | 612 (29.62) | 119 (27.87) | 30 (23.08) | na | 26 (29.55) | 70 (30.84) | 42 (41.58) |
| Alcohol consumption (= yes) | 1,500 (72.60) | 341 (79.86) | 102 (78.46) | na | 64 (72.73) | 158 (69.60) | 66 (65.35) |
| Smoking exposure (= yes) | 276 (13.36) | 64 (14.99) | 22 (16.92) | na | 11 (12.50) | 32 (14.10) | 13 (12.87) |
| Physical activity status | | | | | | | |
| Low | 876 (42.40) | 153 (55.11) | 88 (55.11) | | 36 (55.11) | 98 (55.11) | 52 (55.11) |
| Moderate | 701 (33.93) | 134 (29.55) | 30 (29.55) | | 28 (29.55) | 74 (29.55) | 33 (29.55) |
| High | 489 (23.67) | 140 (15.34) | 12 (15.34) | | 24 (15.34) | 55 (15.34) | 16 (15.34) |
| Health status burden | | | | | | | |
| No burden | 208 (10.07) | 57 (13.35) | 13 (10.00) | na | 24 (27.27) | 19 (8.37) | 1 (0.99) |
| Moderate burden | 1,205 (58.33) | 268 (62.76) | 82 (63.08) | na | 48 (54.55) | 135 (59.47) | 41 (40.59) |
| Severe burden | 653 (31.61) | 102 (23.89) | 35 (26.92) | na | 16 (18.18) | 73 (32.16) | 59 (58.42) |
| Disability status | | | | | | | |
| No disability | 1,172 (56.73) | 258 (60.42) | 76 (58.46) | na | 76 (86.36) | 124 (54.63) | 32 (31.68) |
| Moderate disability | 509 (24.64) | 92 (21.55) | 26 (20.00) | na | 7 (7.95) | 63 (27.75) | 39 (38.61) |
| Severe disability | 385 (18.64) | 77 (18.03) | 28 (21.54) | na | 5 (5.68) | 40 (17.62) | 30 (29.70) |
| Healthcare utilisation (= yes) | 1,093 (72.43) | 219 (65.45) | 63 (46.95) | na | 22 (25.00) | 181 (79.74) | 98 (97.03) |
| Life satisfaction with- | | | | | | | |
| Employment, mean (sd) | 3.39 (3.96) | 3.51 (4.03) | 3.86 (3.94) | na | 5.3 (3.98) | 3.55 (3.98) | 2.36 (3.88) |
| Financial situation, mean (sd) | 6.73 (2.37) | 7.05 (2.27) | 6.65 (2.43) | na | 6.98 (2.14) | 6.63 (2.45) | 6.04 (2.59) |
| Social supports, mean (sd) | 7.83 (1.82) | 8.09 (1.54) | 7.97 (1.54) | na | 7.91 (1.73) | 7.64 (2.03) | 7.78 (1.98) |
| Remoteness | | | | | | | |
| Major Cities | 1,264 (61.18) | 270 (63.23) | 75 (57.69) | na | 48 (54.55) | 128 (56.39) | 63 (62.38) |
| Inner Regional | 519 (25.12) | 98 (22.95) | 34 (26.15) | na | 24 (27.27) | 59 (25.99) | 24 (23.76) |
| Outer Regional | 247 (11.96) | 50 (11.71) | 21 (16.15) | na | 13 (14.77) | 38 (16.74) | 12 (11.88) |
| Remote or very remote | 36 (1.74) | 9 (2.11) | na | na | 3 (3.41) | 2 (0.88) | 2 (1.98) |
| Socioeconomic status | | | | | | | |
| $Q_1$ (lowest 20%) (ref) | 407 (19.70) | 81 (18.97) | 23 (17.69) | na | 11 (12.50) | 46 (20.26) | 27 (26.73) |

*(Continued)*

**Table 1.** (Continued)

| Variables | Number of observations, n (%) | Wave-7 | | | Wave-9 | | |
|---|---|---|---|---|---|---|---|
| | | Number of chronic comorbid conditions, n (%) | | | Number of chronic comorbid conditions, n (%) | | |
| | | 0 | 1–2 | 3 or more | 0 | 1–2 | 3 or more |
| $Q_2$ | 470 (22.75) | 87 (20.37) | 27 (20.77) | na | 16 (18.18) | 60 (26.43) | 29 (28.71) |
| $Q_3$ | 369 (17.86) | 79 (18.50) | 33 (25.38) | na | 25 (28.41) | 39 (17.18) | 14 (13.86) |
| $Q_4$ | 428 (20.72) | 98 (22.95) | 28 (21.54) | na | 20 (22.73) | 39 (17.18) | 21 (20.79) |
| $Q_5$ (highest 20%) | 392 (18.97) | 82 (19.20) | 19 (14.62) | na | 16 (18.18) | 43 (18.94) | 10 (9.90) |
| Overall | 2,066 (100) | 427 (76.66) | 130 (23.34) | na | 88 (21.15) | 227 (54.57) | 101 (24.28) |

Na = not available

confidence interval, *CI*: 1.08, 1.45) compared with a young patient (<25 years). Patients who performed lower levels of physical activity were 1.25 times more likely to have a chronic comorbid condition (*IRR* = 1.25; 95% CI: 1.09, 1.59) compared with patients who engaged in high-level physical activity. Further, patients who faced an extreme health burden were 2.30 times significantly higher risk of having comorbid conditions than those with no health burden. The risks of having a comorbid condition were more pronounced among patients who suffered from extreme health burden (*IRR* = 2.30 times) or moderate burden level (*IRR* = 1.90 times) compared with patients who reported excellent health status. Similarly, a higher risk of having a comorbid exposure was significantly observed in cancer patients who lived in the poorest households (*IRR* = 1.21; 95% CI: 1.11, 1.29) compared with their richest counterparts.

## Discussion

The study results show that approximately 63% of cancer patients suffered from at least one chronic disease. The most prevalent comorbid conditions were arthritis or osteoporosis, high blood pressure or hypertension, obesity, depression or anxiety, heart disease, and asthma. However, these were significantly increased in the presence of diabetes, depression or anxiety, mental illness, heart disease and obesity over time. In the adjusted model, older patients, inadequate level of physical activities, the magnitude of health burden associated with cancer, utilisation of healthcare, and patients living in the poorest households were significant predictors associated with a higher risk of comorbid conditions.

Further, patients who faced an extreme health burden had a three times higher risk of having comorbid conditions than who reported excellent health status. Some studies have confirmed that the poor health status of cancer patients resulted in a greater burden of functional disability (e.g., specific task difficulties) [60,61] along with a higher burden of chronic diseases [15,30,62]. However, the prevalence of long-term health problems, including chronic illness, short or long-term disability, was also more concentrated in combination with a cancer diagnosis [63–68]. Advanced cancer treatments can damage healthy cells or organs [69], for example, radiation and chemotherapy may impose short and long-term chronic health problems and impact on the spinal cord, nerves, and brain, which then may significantly contribute to long-term adverse health outcomes like death, physical and mental disabilities.

The results indicate that aged cancer patients (older than 65 years) were at a 1.15 times higher risk of having chronic comorbid conditions compared with younger patients. This finding is consistent with previous studies, which revealed that elderly cancer patients reported significantly more exposure to chronic comorbid conditions [23,70,71], required more assistance with daily living activities [72], and had deficits in performing work-related activities in terms

**Table 2. Summary statistics by the number of chronic condition among cancer patients for wave 13 and wave 17.**

| Variables | Wave-13 | | | Wave-17 | | | Overall | | |
|---|---|---|---|---|---|---|---|---|---|
| | Number of chronic comorbid conditions, n (%) | | | Number of chronic comorbid conditions, n (%) | | | Number of chronic comorbid conditions, n(% | | |
| | 0 | 1–2 | 3 or more | 0 | 1–2 | 3 or more | 0 | 1–2 | 3 or more |
| Sex | | | | | | | | | |
| *Male* | 70 (58.82) | 122 (50.41) | 87 (55.77) | 73 (57.94) | 160 (58.39) | 90 (51.14) | 422 (55.53) | 469 (53.72) | 232 (53.58) |
| *Female* | 49 (41.18) | 120 (49.59) | 69 (44.23) | 53 (42.06) | 114 (41.61) | 86 (48.86) | 338 (44.47) | 404 (46.28) | 201 (46.42) |
| Age | | | | | | | | | |
| *<25 years* | 6 (5.04) | 9 (3.72) | 2 (1.28) | 6 (4.76) | 5 (1.82) | 6 (3.41) | 23 (3.03) | 21 (2.41) | 9 (2.08) |
| *25–45 years* | 25 (21.01) | 30 (12.4) | 9 (5.77) | 26 (20.63) | 28 (10.22) | 8 (4.55) | 151 (19.87) | 107 (12.26) | 25 (5.77) |
| *46–65 years* | 51 (42.86) | 93 (38.43) | 47 (30.13) | 56 (44.44) | 98 (35.77) | 56 (31.82) | 292 (38.42) | 346 (39.63) | 133 (30.72) |
| *>65 years* | 37 (31.09) | 110 (45.45) | 98 (62.82) | 38 (30.16) | 143 (52.19) | 106 (60.23) | 294 (38.68) | 399 (45.7) | 266 (61.43) |
| Educational attainment | | | | | | | | | |
| *Year 11 or below* | 31 (26.05) | 88 (36.36) | 70 (44.87) | 30 (23.81) | 91 (33.21) | 78 (44.32) | 256 (33.68) | 324 (37.11) | 194 (44.8) |
| *Year 12* | 9 (7.56) | 20 (8.26) | 8 (5.13) | 12 (9.52) | 21 (7.66) | 13 (7.39) | 68 (8.95) | 70 (8.02) | 30 (6.93) |
| *Trade/certificate/diploma* | 51 (42.86) | 107 (44.21) | 59 (37.82) | 47 (37.3) | 117 (42.7) | 68 (38.64) | 282 (37.11) | 359 (41.12) | 167 (38.57) |
| *Tertiary* | 28 (23.53) | 27 (11.16) | 19 (12.18) | 37 (29.37) | 45 (16.42) | 17 (9.66) | 154 (20.26) | 120 (13.75) | 42 (9.7) |
| Unemployed | 58 (48.74) | 159 (65.70) | 128 (82.05) | 54 (42.86) | 177 (64.60) | 138 (78.41) | 402 (52.89) | 552 (63.23) | 352 (81.29) |
| Marital status | | | | | | | | | |
| *Single* | 21 (17.65) | 30 (12.4) | 12 (7.69) | 24 (19.05) | 30 (10.95) | 19 (10.8) | 113 (14.87) | 107 (12.26) | 38 (8.78) |
| *Married* | 72 (60.5) | 141 (58.26) | 90 (57.69) | 69 (54.76) | 164 (59.85) | 96 (54.55) | 443 (58.29) | 515 (58.99) | 238 (54.97) |
| *Others* | 26 (21.85) | 71 (29.34) | 54 (34.62) | 33 (26.19) | 80 (29.2) | 61 (34.66) | 204 (26.84) | 251 (28.75) | 157 (36.26) |
| Alcohol consumption (= yes) | 91 (76.47) | 178 (73.55) | 100 (64.10) | 84 (66.67) | 205 (74.82) | 111 (63.07) | 580 (76.32) | 643 (73.65) | 277 (63.97) |
| Smoking exposure (= yes) | 11 (9.24) | 32 (13.22) | 23 (14.74) | 14 (11.11) | 32 (11.68) | 22 (12.50) | 100 (13.16) | 118 (13.52) | 58 (13.39) |
| Physical activity status | | | | | | | | | |
| *Low* | 35 (29.41) | 95 (39.26) | 94 (60.26) | 50 (39.68) | 125 (45.62) | 97 (55.11) | 274 (36.05) | 406 (46.51) | 243 (56.12) |
| *Moderate* | 44 (36.97) | 81 (33.47) | 44 (28.21) | 36 (28.57) | 89 (32.48) | 52 (29.55) | 242 (31.84) | 274 (31.39) | 129 (29.79) |
| *High* | 40 (33.61) | 66 (27.27) | 18 (11.54) | 40 (31.75) | 60 (21.9) | 27 (15.34) | 244 (32.11) | 193 (22.11) | 61 (14.09) |
| Health status burden | | | | | | | | | |
| *No burden* | 30 (25.21) | 15 (6.2) | 0 (0) | 22 (17.46) | 22 (8.03) | 5 (2.84) | 132 (17.37) | 69 (7.9) | 6 (1.39) |
| *Moderate burden* | 64 (53.78) | 172 (71.07) | 75 (48.08) | 76 (60.32) | 175 (63.87) | 69 (39.2) | 422 (55.53) | 513 (58.76) | 156 (36.03) |
| *Severe burden* | 25 (21.01) | 55 (22.73) | 81 (51.92) | 28 (22.22) | 77 (28.1) | 102 (57.95) | 206 (27.11) | 291 (33.33) | 271 (62.59) |
| Disability status | | | | | | | | | |
| *No disability* | 96 (80.67) | 146 (60.33) | 41 (26.28) | 104 (82.54) | 153 (55.84) | 66 (37.50) | 534 (70.26) | 499 (57.16) | 139 (32.10) |
| *Moderate disability* | 9 (7.56) | 59 (24.38) | 56 (35.9) | 10 (7.94) | 84 (30.66) | 64 (36.36) | 118 (15.53) | 232 (26.58) | 159 (36.72) |
| *Severe disability* | 14 (11.76) | 37 (15.29) | 59 (37.82) | 12 (9.52) | 37 (13.5) | 46 (26.14) | 108 (14.21) | 142 (16.27) | 135 (31.18) |
| Healthcare utilisation (= yes) | 39 (32.77) | 175 (72.31) | 152 (97.44) | 47 (37.30) | 209 (76.28) | 170 (96.59) | 108 (9.88) | 565 (51.69) | 420 (38.43) |
| Life satisfaction with- | | | | | | | | | |
| *Employment, mean (sd)* | 4.82 (3.89) | 3.47 (3.9) | 1.96 (3.43) | 4.48 (3.97) | 3.39 (3.97) | 1.64 (3.12) | 4.08 (4.04) | 3.52 (3.95) | 1.92 (3.43) |
| *Financial situation, mean (sd)* | 7.39 (1.98) | 6.5 (2.53) | 5.99 (2.57) | 7.33 (2.01) | 6.76 (2.31) | 6.32 (2.65) | 7.14 (2.17) | 6.64 (2.43) | 6.13 (2.6) |
| *Social supports, mean (sd)* | 7.94 (1.68) | 7.67 (2.17) | 7.74 (1.95) | 7.82 (1.74) | 7.92 (1.71) | 7.44 (2.1) | 8 (1.62) | 7.78 (1.91) | 7.63 (2.02) |
| Remoteness | | | | | | | | | |
| *Major Cities* | 79 (66.39) | 151 (62.4) | 105 (67.31) | 91 (72.22) | 151 (55.11) | 103 (58.52) | 488 (64.21) | 505 (57.85) | 271 (62.59) |
| *Inner Regional* | 27 (22.69) | 55 (22.73) | 35 (22.44) | 26 (20.63) | 87 (31.75) | 50 (28.41) | 175 (23.03) | 235 (26.92) | 109 (25.17) |
| *Outer Regional* | 11 (9.24) | 32 (13.22) | 16 (10.26) | 6 (4.76) | 28 (10.22) | 20 (11.36) | 80 (10.53) | 119 (13.63) | 48 (11.09) |
| *Remote or very remote* | 2 (1.68) | 4 (1.65) | 0 (0) | 3 (2.38) | 8 (2.92) | 3 (1.7) | 17 (2.24) | 14 (1.6) | 5 (1.15) |
| Socioeconomic status | | | | | | | | | |
| $Q_1$ *(lowest 20%) (ref)* | 17 (14.29) | 50 (20.66) | 48 (30.77) | 18 (14.29) | 41 (14.96) | 45 (25.57) | 127 (16.71) | 160 (18.33) | 120 (27.71) |

*(Continued)*

**Table 2.** (Continued)

| Variables | Wave-13 | | | Wave-17 | | | Overall | | |
|---|---|---|---|---|---|---|---|---|---|
| | Number of chronic comorbid conditions, n (%) | | | Number of chronic comorbid conditions, n (%) | | | Number of chronic comorbid conditions, n(% | | |
| | 0 | 1–2 | 3 or more | 0 | 1–2 | 3 or more | 0 | 1–2 | 3 or more |
| $Q_2$ | 22 (18.49) | 59 (24.38) | 38 (24.36) | 22 (17.46) | 62 (22.63) | 48 (27.27) | 147 (19.34) | 208 (23.83) | 115 (26.56) |
| $Q_3$ | 21 (17.65) | 29 (11.98) | 27 (17.31) | 21 (16.67) | 51 (18.61) | 30 (17.05) | 146 (19.21) | 152 (17.41) | 71 (16.40) |
| $Q_4$ | 32 (26.89) | 51 (21.07) | 23 (14.74) | 27 (21.43) | 64 (23.36) | 25 (14.20) | 177 (23.29) | 182 (20.85) | 69 (15.94) |
| $Q_5$ *(highest 20%)* | 27 (22.69) | 53 (21.9) | 20 (12.82) | 38 (30.16) | 56 (20.44) | 28 (15.91) | 163 (21.45) | 171 (19.59) | 58 (13.39) |
| *Overall* | 119 (23.02) | 242 (46.81) | 156 (30.17) | 126 (21.88) | 274 (47.57) | 176 (30.56) | 760 (36.79) | 873 (42.26) | 433 (20.96) |

Na = not available

of their physical ability [60,73]. Several reasons might influence this reduction in their physical strength. For example, a course of advanced cancer treatment is associated with considerable physical and psychological side effects in elderly cancer patients (e.g., weight change, muscle loss, fatigue, and physical weakness) [74], and exposure to multiple comorbidities [64,65,75] will presumably contribute to worse health status. Although, cancer patients in older age groups are less likely to be offered cancer treatments (e.g., chemotherapy, radiotherapy and axillary lymph node dissection) that may then contribute to a greater burden of health [74]. This result indicates that rehabilitation-related interventions (e.g., physical therapies) are essential to prevent or alleviate chronic comorbid conditions and an emerging cancer research area, particularly focused on the elderly [76].

The present study found that cancer patients who performed lower levels of physical activities were strongly associated with an extreme level of chronic comorbidities compared with patients engaged in high-level physical activity. This finding is in line with other studies [52,77,78], whereby it was found that limited physical activity levels were significantly associated with a higher risk of having chronic comorbid conditions in cancer patients. The magnitude of limited physical activity level may decrease the risk for several cancers by some mechanisms, including decreasing sex hormones, metabolic hormones and inflammation, and improving immune function [77]. In terms of cancer risk, high levels of physical activities (compared with low levels) played a significant role in the prevention of several cancers (e.g., 42% for gastrointestinal cancer, 23% for renal cancer, and 20% for myeloid leukemia) [79].

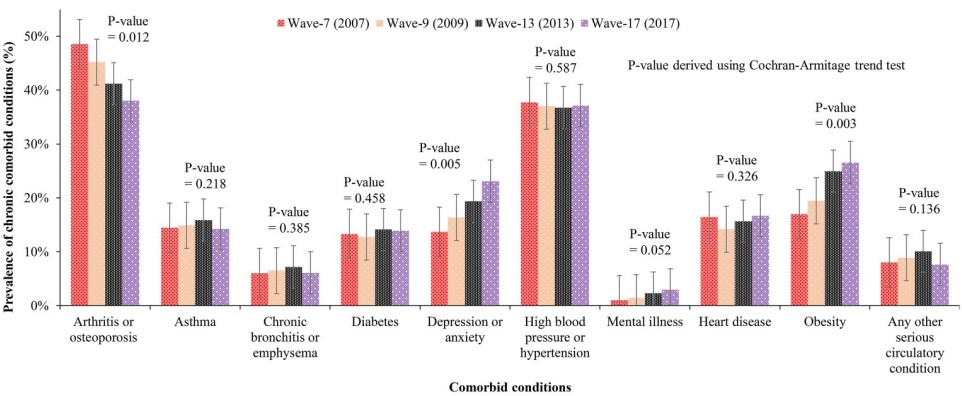

**Fig 2. The trend of disease pattern among patients with cancer.**

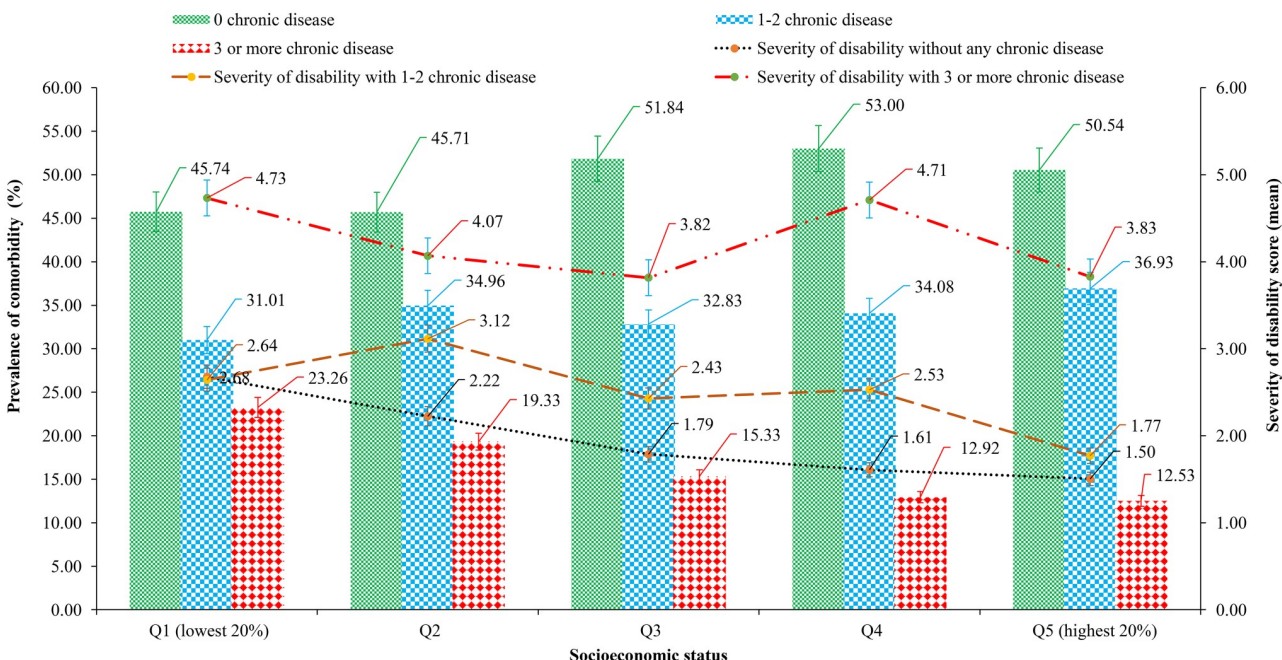

**Fig 3. Unequal distribution of the presence of chronic comorbidities with the severity of disability among cancer patients across socioeconomic status.**

This includes averting genetic damage, improving the immune system, reducing chronic infections, and controlling cancer cells [79]. In addition, some past studies confirmed that physical activity plays an effective role in controlling the side effects of cancer treatment and disease progression, reducing psychological conditions [77,80] and reducing the risk of developing future cancers [81]. Several hypotheses and mechanisms have been suggested regarding the anti-cancer effects of physical activities. The American Cancer Society guidelines for cancer survivors [82] recommend daily physical activities, including a continuation of normal daily life activities immediately after diagnosis, which help to significantly reduce physical stamina and muscle strength erosion as well as anxiety levels, thereby resulting in the prevention of long-term adverse health outcomes (e.g., extreme comorbidity burden and disability) [83]. In this context, future research could examine the influence that physical activity has on the effectiveness of chronic comorbid conditions among cancer patients.

The risks of having extreme chronic comorbidity conditions amongst cancer patients who lived in the poorest households were more pronounced compared with their richer counterparts. Recent studies confirm this result with the disadvantaged socioeconomic status of cancer survivors being negatively associated with long-term adverse health outcomes (e.g., multiple chronic illnesses, physical disability) [83–93]. Some studies also provided evidence that the magnitude of the cancer burden is adversely associated with socioeconomic status [16, 32–35]. Further, adverse cancer outcomes (e.g., worse health status and long-term chronic illness) were disproportionately found in poorer people as opposed to those of higher socioeconomic status [13, 16, 32, 34]. Some reasons that have contributed to the high rates of long term health impacts among the poorest groups include higher tobacco consumption [16,28], economic burden [36,37], increased mental illness [94], lack of health education and awareness [95], and less access to competent and effective health care services [95]. Low productivity, loss/reduction of household income, and increased healthcare expenditure are more pronounced amongst the poorest cancer patients. Growing socioeconomic disparities of cancer outcomes need the

**Table 3. Factors influencing chronic comorbid conditions of cancer patients using a fixed-effect negative binomial regression model.**

| Variables | Unadjusted model[1] | | Adjusted model[2] | |
|---|---|---|---|---|
| | IRR (SE) | 95% CI | IRR (SE) | 95% CI |
| Female (ref = male) | 1.04 (0.05) | (0.94, 1.14) | - | - |
| Age group | | | | |
| < 25 years (= ref) | 1.00 | - | 1.00 | - |
| 25–45 years | 0.72 (0.13) | (0.51, 1.03) | 0.85 (0.14) | (0.61, 1.18) |
| 46–65 years | 1.15 (0.19) | (0.83, 1.58) | 1.07 (0.16) | (0.79, 1.45) |
| >65 years | 1.49*** (0.24) | (1.09, 2.04) | 1.15** (0.17) | (1.08, 1.45) |
| Educational attainment | | | | |
| Year 11 or below | 1.48*** (0.12) | (1.26, 1.74) | 1.16** (0.09) | (1.01, 1.35) |
| Year 12 | 1.11 (0.13) | (0.88, 1.40) | 1.13 (0.12) | (0.91, 1.40) |
| Trade/certificate/diploma | 1.38*** (0.12) | (1.17, 1.63) | 1.21*** (0.09) | (1.05, 1.40) |
| Tertiary (= ref) | 1.00 | - | 1.00 | - |
| Unemployed (ref = employed) | 1.80*** (0.10) | (1.62, 2.00) | 1.08 (0.07) | (0.95, 1.23) |
| Marital status | | | | |
| Single (= ref) | 1.00 | - | 1.00 | - |
| Married | 1.21** (0.10) | (1.02, 1.42) | 1.02 (0.08) | (0.87, 1.20) |
| Others | 1.41*** (0.12) | (1.19, 1.68) | 1.06 (0.09) | (0.90, 1.25) |
| Physical activity status | | | | |
| Low | 1.60*** (0.12) | (1.39, 1.85) | 1.25** (0.07) | (1.09, 1.59) |
| Moderate | 1.30*** (0.10) | (1.12, 1.52) | 1.06 (0.07) | (0.92, 1.21) |
| High (= ref) | 1.00 | - | 1.00 | - |
| Alcohol consumption (ref = yes) | 1.26*** (0.06) | (1.14, 1.39) | 0.91 (0.05) | (0.82, 1.00) |
| Smoking exposure (ref = no) | 1.02 (0.07) | (0.88, 1.18) | - | - |
| Healthcare utilisation (ref = no) | 0.27 (0.02) | (0.24, 0.31) | 0.38*** (0.03) | (0.33, 0.45) |
| Health status burden | | | | |
| No burden (= ref) | 1.00 | - | 1.00 | - |
| Moderate burden | 2.44*** (0.27) | (1.96, 3.03) | 1.90*** (0.26) | (1.45, 2.48) |
| Severe burden | 4.18*** (0.47) | (3.36, 5.21) | 2.30*** (0.33) | (1.73, 3.05) |
| Disability status | | | | |
| No disability (= ref) | 1.00 | - | 1.00 | - |
| Moderate disability | 1.82*** (0.10) | (1.64, 2.02) | 1.22*** (0.07) | (1.10, 1.36) |
| Severe disability | 1.99*** (0.12) | (1.76, 2.24) | 1.25*** (0.08) | (1.11, 1.41) |
| Life satisfaction with- | | | | |
| Employment | 0.94*** (0.01) | (0.92, 0.95) | 0.98*** (0.01) | (0.97, 0.99) |
| Financial situation | 0.97** (0.01) | (0.95, 0.99) | 0.96*** (0.01) | (0.94, 0.98) |
| Social supports | 0.96*** (0.01) | (0.93, 0.98) | 1.03** (0.01) | (1.01, 1.05) |
| Remoteness | | | | |
| Major cities (= ref) | 1.00 | - | - | - |
| Inner regional | 1.02 (0.06) | (0.91, 1.14) | - | - |
| Outer regional | 1.04 (0.08) | (0.90, 1.21) | - | - |
| Remote or very remote | 0.77 (0.14) | (0.54, 1.11) | - | - |
| Socioeconomic status | | | | |
| $Q_1$ (lowest 20%) | 1.51*** (0.12) | (1.29, 1.77) | 1.21*** (0.08) | (1.11, 1.29) |
| $Q_2$ | 1.35*** (0.11) | (1.15, 1.57) | 1.09 (0.08) | (0.95, 1.26) |
| $Q_3$ | 1.19** (0.10) | (1.01, 1.41) | 1.15 (0.09) | (0.99, 1.34) |
| $Q_4$ | 1.08 (0.09) | (0.92, 1.27) | 0.99 (0.08) | (0.85, 1.15) |

*(Continued)*

**Table 3.** (Continued)

| Variables | Unadjusted model[1] | | Adjusted model[2] | |
|---|---|---|---|---|
| | IRR (SE) | 95% CI | IRR (SE) | 95% CI |
| $Q_5$ (highest 20%) (= ref) | 1.00 | - | 1.00 | - |

Note:

[*]p<0.05,

[**]p<0.01,

[***]p<0.001,

*IRR* = incidence rate ratio, *SE* = standard error, *CI* = confidence interval,

[1]Single explanatory variable was included in un-adjusted model,

[2]Explanatory variables were included in the adjusted model only if any label of the variable was significant at 5% or less risk level in the unadjusted model

attention of governments, health systems, and decision-makers. For example, Cancer Council Australia has an optimal care pathway project, which has already addressed several cancer sites in disadvantaged areas. Such initiatives might help to reduce socio-economic disparities, which are related to poverty, gender, education, and health, and they should promote universal access to health care which can further enhance both socio-economic and human development.

This study has some limitations. Study participants were accessed from the *HILDA* survey, which covers health, economic, employment, income and health characteristics of household members aged 15 years and older. Children who suffered from cancer were excluded from this study. The study findings established a relationship between cancer diagnosis and chronic comorbidity conditions among cancer survivors, which might vary in terms of cancer stages and types of cancer. The authors were not able to estimate the cancer type analysis due to the paucity of relevant data. Further, the study findings were based on self-reported responses that might have been impacted by respondents' prejudice (e.g., silence and over-response), and by problems in understanding and interpreting the survey questions.

Despite these limitations, this study has some strengths including the use of a prospective longitudinal design of long term follow-ups and the application of well-validated and reliable longitudinal wave measures of the impacts of a cancer diagnosis on the burden of chronic comorbid conditions of individuals over the 2007–2017 period. The study population captured different dimensions including ethnically, geographically, and socio-economically diverse groups. Furthermore, this study included several potential confounding factors such as health status burden, the severity of the disability level as well as life satisfaction (e.g., employment, financial situation and, social supports) that were not present in previous studies. For this study, data were gathered from four-wave of the *HILDA* survey for cancer survivors. The length of the survey period may have introduced uncontrolled bias, as changes in health status are not instantaneous and might emerge only after time, which was not captured in this study. Due to the paucity of funding, the authors were unable to consider cancer patients who registered for cancer surveillance as well as received health care from other health facilities (e.g., private clinics, community clinics and, secondary or tertiary hospitals). Future study is required using a similar study design, perspective, and analytical methods in terms of cancer-specific exploration.

## Conclusions

This study has shown an extreme burden of chronic comorbid conditions among cancer patients in Australia. Older patients, inadequate level of physical activities, the magnitude of health burden, and patients living in the poorest households were significant predictors associated with a higher risk of having chronic comorbidity conditions. The findings have further

implications for improving public health policy and reducing population-level unhealthy life-styles, which should be recommended. The study results could be used to better outline the management of a sequelae course of treatment for those who should undergo more intensive physical rehabilitation aimed at reducing the risk of adverse health outcomes. Given the clinical significance of comorbidity in cancer survivors, this study may play a significant role in providing comprehensive evidence for health care providers, including physical therapists and oncologists, who should be aware of the unique problems that challenge this population and who should advocate for prevention and evidence-based interventions. Finally, a greater awareness of the importance of managing a patients overall health status within the context of comorbidity is warranted together with emphasised research on comorbidity to generate an appropriate scientific basis on which to build evidence-based care guidelines for these chronic comorbid conditions patients.

## Acknowledgments

The study is part of the first author's PhD research at the University of Southern Queensland, Australia. We would also like to thank the Australian Government's Department of Social Services (*DSS*), the *HILDA* study at Melbourne Institute for providing access to the data used in the research. We would like to gratefully acknowledge the study participants, reviewers, and editors of our manuscript.

## Author Contributions

**Conceptualization:** Rashidul Alam Mahumud, Khorshed Alam, Jeff Dunn, Jeff Gow.

**Data curation:** Rashidul Alam Mahumud.

**Formal analysis:** Rashidul Alam Mahumud.

**Funding acquisition:** Khorshed Alam, Jeff Dunn, Jeff Gow.

**Investigation:** Rashidul Alam Mahumud, Khorshed Alam, Jeff Dunn.

**Methodology:** Rashidul Alam Mahumud.

**Project administration:** Rashidul Alam Mahumud, Khorshed Alam, Jeff Dunn, Jeff Gow.

**Resources:** Rashidul Alam Mahumud, Khorshed Alam, Jeff Dunn, Jeff Gow.

**Software:** Rashidul Alam Mahumud.

**Supervision:** Khorshed Alam, Jeff Dunn, Jeff Gow.

**Validation:** Rashidul Alam Mahumud, Khorshed Alam, Jeff Dunn.

**Visualization:** Rashidul Alam Mahumud.

**Writing – original draft:** Jeff Dunn, Jeff Gow.

**Writing – review & editing:** Rashidul Alam Mahumud, Khorshed Alam, Jeff Dunn, Jeff Gow.

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
