## [Decision Letter · Decision Letter 0]

14 Oct 2019

PONE-D-19-24164

The emerging burden of chronic diseases among Australian cancer patients: Evidence from a longitudinal exploration, 2003-2017

PLOS ONE

Dear Mr. Mahumud,

Thank you for submitting your manuscript to PLOS ONE. After careful consideration, we feel that it has merit but does not fully meet PLOS ONE’s publication criteria as it currently stands. Therefore, we invite you to submit a revised version of the manuscript that addresses the points raised during the review process.

We would appreciate receiving your revised manuscript by Nov 28 2019 11:59PM. To enhance the reproducibility of your results, we recommend that if applicable you deposit your laboratory protocols in protocols.io, where a protocol can be assigned its own identifier (DOI) such that it can be cited independently in the future. For instructions see: http://journals.plos.org/plosone/s/submission-guidelines#loc-laboratory-protocols

We look forward to receiving your revised manuscript.

Kind regards,

Miguel Angel Luque-Fernandez

Academic Editor

PLOS ONE

Journal Requirements:

3. Please include your tables as part of your main manuscript and remove the individual files. Please note that supplementary tables (should remain/ be uploaded) as separate "supporting information" files

Reviewers' comments:

Reviewer's Responses to Questions

**Comments to the Author**

1. Is the manuscript technically sound, and do the data support the conclusions?

Reviewer #1: Yes

Reviewer #2: Partly

2. Has the statistical analysis been performed appropriately and rigorously? 

Reviewer #1: Yes

Reviewer #2: No

3. Have the authors made all data underlying the findings in their manuscript fully available?

Reviewer #1: Yes

Reviewer #2: Yes

4. Is the manuscript presented in an intelligible fashion and written in standard English?

Reviewer #1: Yes

Reviewer #2: Yes

5. Review Comments to the Author

Reviewer #1: Thank you for the opportunity to read and comment on your manuscript, this is an interesting study. I have some comments and suggestions, which I list below.

1) The title is quite broad, and you describe the burden as “emerging”. I’m not clear if by ‘burden’ you are referring to the presence of chronic disease in cancer patients (and the complexity this comorbidity presents in the care of cancer patients), and what the evidence is from your study to suggest this is an emerging burden?

2) I would have liked more detail about the cancer patient population in this study, specifically the timing of when the cancer had been diagnosed (in relation to the timing of the survey) and the type of cancer diagnosed. The latter in particular is really important in relation to this study, given that the aetiology of cancers can vary according to their type, and that some cancers share common risk factors with some of the chronic conditions you investigated (e.g. smoking is strongly associated with both lung cancer and emphysema). In addition, was the distribution of cancer types similar across each wave of the study? Are the changes in chronic comorbid conditions over time presented in Figure 1 a like-for-like comparison among the cancer patient populations?

3) Explanatory variables.

i) Is there a main exposure of interest?

ii) Why did you include the variable 'satisfaction with household members' as a predictor in your analysis, how does this relate to comorbidity in cancer patients? Was it considered to be a proxy variable for stress?! A conceptual framework diagram may help to explain your assumptions of the relationships between your explanatory variables and outcome variable.

4) Table 1: it would be interesting to include a summary of the distribution of the number of chronic conditions according to the wave of the study.

Minor comments

5) Avoid vague statements, specifically:

i) Top of page 4: "Existing research recognises the critical role played by comorbid chronic conditions among cancer patients".

ii) Page 5: "Individuals with cancer who faced the burden of chronic comorbid conditions were investigated to see the degree of the cancer burden related to its primary evaluation as well as their ability to cope". What do you mean by the degree of the cancer burden relating to its primary evaluation? How are you quantifying ability to cope?

Reviewer #2: The present manuscript titled ‘The emerging burden of chronic diseases among Australian cancer patients: Evidence from a longitudinal exploration, 2003-2017' presents a prospective longitudinal design using data from the HILDA Australian survey. Authors have applied a fixed-effect negative binomial regression model to predict the potential factors of occurrence in chronic comorbid conditions. In general, the theme of the manuscript is important and relevant. However, I found some key points unexplained or not covered in the present form of the manuscript. I would suggest a revision (given below) before the acceptance of the manuscript for publication in the PloS One.

My suggestions/comments are as follows:

(1) Introduction: The Introduction provides information concerning cancer and comorbid conditions worldwide, but does not provide enough information regarding the Australian reality. I suggest the authors explain better how this study is important to the cancer epidemiology field, specifying the challenges, particularly to Australia.

(2) Methods:

Data source: I suggest the authors provide more information concerning HILDA Australian survey. In the text, the authors have described information for five waves but this information is unclear when we analyze the figures and tables. It is unclear why the authors selected five waves with cancer-related information but have just presented data for three waves in Figure 1.

Outcome variable: The authors have cited that “there is no gold-standard method to measuring comorbidity status in the context of cancer patients”, then the authors have listed some methods for measuring comorbidity status applied to the literature. As a reader, I would like to know why the authors have chosen the count of comorbid conditions? It was a single count? How this information was organized? Please, provide more information explaining these details.

Explanatory variables: The authors applied several demographic, socio-economic and health and lifestyle-related variables in this study. For each variable, there is a specific categorization method. For some variables the rationale of applying the scale is unclear, an example is the application of the SF-36 scale. I know what the survey means but it is necessary to explain it in the text. I didn’t get the rationale of why the authors have chosen this quality of life scale to justify health burden levels and also how the score was made in this study. I also didn’t see any information as supplementary material of these data. Please, I suggest the authors clarify them.

Statistical analysis: What the authors understand as an “insignificant predictors were not included in the adjusted model”? Please, provide more information and explain it. I understand that the adjusted model was applied as a fixed-effect negative binomial regression. But what method was applied and defined as unadjusted? I suggest again the authors clarify and explain better this important information for future readers of this manuscript.

(3) Results:

- The quality of the Figures is not good. Please, provide the Figures in better quality. I simply cannot read the legend of Figure 3.

- Why the authors have only shown information for three waves in Figure 1 if in the Methods section were described five waves? The authors have applied a Cochran- Armitage trend test. It was the rationale of present the results for three waves? Please, explain it.

- As a reader, It will be more interesting to see in Table 2 the results for each comorbid condition such as the authors have shown in Table 1 (0 chronic diseases; 1-2 chronic diseases; 3 or more chronic diseases).

- I didn’t see any information as supplementary material of all the several variables analyzed in this study.

(4) Discussion:

Your discussion is interesting and you made an effort to compare your results with the results of previous studies. However, what is your specific recommendations based on the results you produced to Australian reality? Please, provide more information on what has been studied in Australia and why your study is important on this topic.

(4) Minor comments:

Some sentences do not read well. Please consider revising.

6. PLOS authors have the option to publish the peer review history of their article (what does this mean?). If published, this will include your full peer review and any attached files.

Reviewer #1: No

Reviewer #2: No

---

## [Author Response · Author response to Decision Letter 0]

12 Dec 2019

Dear Reviwer(s),

Thank you for giving us an opportunity to revise our manuscript entitled “The burden of chronic diseases among Australian cancer patients: Evidence from a longitudinal exploration, 2007-2017”. We found the reviewers’ comments/feedback very helpful in improving the manuscript and we have revised the manuscript accordingly. Please find attached the revised manuscript. We have no conflict of interest to declare. The manuscript has not been published in any other journal. Our point-by-point comments on the suggested revisions are below.

---

## [Decision Letter · Decision Letter 1]

14 Jan 2020

PONE-D-19-24164R1

The burden of chronic diseases among Australian cancer patients: Evidence from a longitudinal exploration, 2007-2017

PLOS ONE

Dear Mr. Mahumud,

Thank you for submitting your manuscript to PLOS ONE. We would like to accept for publication your article but there is still a couple of minor questions raised by one reviewer that you would like to answer/clarify. Therefore, we invite you to submit a revised version of the manuscript that addresses the points raised during the review process.

We would appreciate receiving your revised manuscript by Feb 28 2020 11:59PM. To enhance the reproducibility of your results, we recommend that if applicable you deposit your laboratory protocols in protocols.io, where a protocol can be assigned its own identifier (DOI) such that it can be cited independently in the future. For instructions see: http://journals.plos.org/plosone/s/submission-guidelines#loc-laboratory-protocols

We look forward to receiving your revised manuscript.

Kind regards,

Miguel Angel Luque-Fernandez

Academic Editor

PLOS ONE

Reviewers' comments:

Reviewer's Responses to Questions

**Comments to the Author**

1. If the authors have adequately addressed your comments raised in a previous round of review and you feel that this manuscript is now acceptable for publication, you may indicate that here to bypass the “Comments to the Author” section, enter your conflict of interest statement in the “Confidential to Editor” section, and submit your "Accept" recommendation.

Reviewer #1: All comments have been addressed

Reviewer #2: All comments have been addressed

2. Is the manuscript technically sound, and do the data support the conclusions?

Reviewer #1: Yes

Reviewer #2: Yes

3. Has the statistical analysis been performed appropriately and rigorously? 

Reviewer #1: Yes

Reviewer #2: Yes

4. Have the authors made all data underlying the findings in their manuscript fully available?

Reviewer #1: No

Reviewer #2: Yes

5. Is the manuscript presented in an intelligible fashion and written in standard English?

Reviewer #1: Yes

Reviewer #2: Yes

6. Review Comments to the Author

Reviewer #1: Thank you for the opportunity to read the revised version of your manuscript.

I have a couple of further comments:

(Using your numbering system of the original comments - comment 3: (page 18, lines 26-29) you have used the word ‘emerging’ in this sentence, please clarify what this means. For example, are you saying that the prevalence of comorbidity among cancer patients has increased over time? I think you need to be more explicit. Likewise the sentence “The authors were not able to estimate the cancer-specific analysis due to the paucity of relevant data” – what do you mean by cancer-specific? According to cancer type?

Reviewer #2: This is the second version of the manuscript. Authors have done considerable changes explaining in detail point-by-point according to peer reviewer's suggestions/comments and it is commendable. The revision certainly improves the quality and scope of the present manuscript. I would recommend it for possible publication in PlosOne.

7. PLOS authors have the option to publish the peer review history of their article (what does this mean?). If published, this will include your full peer review and any attached files.

Reviewer #1: No

Reviewer #2: No

---

## [Author Response · Author response to Decision Letter 1]

22 Jan 2020

Please find the attached file. We found the reviewers’ comments/feedback very helpful in improving the manuscript and we have revised the manuscript accordingly.

---

## [Editor Report · Decision Letter 2]

23 Jan 2020

The burden of chronic diseases among Australian cancer patients: Evidence from a longitudinal exploration, 2007-2017

PONE-D-19-24164R2

Dear Dr. Mahumud,

We are pleased to inform you that your manuscript has been judged scientifically suitable for publication and will be formally accepted for publication once it complies with all outstanding technical requirements.

With kind regards,

Miguel Angel Luque-Fernandez

Academic Editor

PLOS ONE
---

## [Editor Report · Acceptance letter]

31 Jan 2020

PONE-D-19-24164R2 

The burden of chronic diseases among Australian cancer patients: Evidence from a longitudinal exploration, 2007-2017 

Dear Dr. Mahumud:

I am pleased to inform you that your manuscript has been deemed suitable for publication in PLOS ONE. Congratulations! Your manuscript is now with our production department. 

With kind regards,

on behalf of

Dr. Miguel Angel Luque-Fernandez 

Academic Editor

PLOS ONE